# SarNet: Sarcasm vs True-Hate Detection Network

## Abstract

At times hate speech detection classifiers miss the context of a sentence and flag a sarcastic tweet incorrectly. To tackle this problem by emphasising on the context of a tweet we propose SarNet. SarNet is a two-fold deep learning based model which follows a quasi-ternary labelling strategy and contextually classifies a tweet as hate, sarcastic or neither. The first module of SarNet is an ANN-BiLSTM based Pyramid Network used to calculate the hate and sarcastic probabilities of a sentence. The second module of the SarNet is the Nash Equalizer which stems from the concept of game theory and prisoner's dilemma. It treats hate and sarcasm as two prisoners. A payoff matrix is constructed to calculate the true hate of the tweet. True hate considers the hate part of a tweet excluding the sarcastic part of the tweet. Thus, this gives a true estimate of the hate content in a tweet thereby decreasing the number of sarcastic tweets being falsely flagged as hate. Our proposed model is trained on state-of-the-art hate speech and sarcasm datasets in the English language. The precision, recall and F1 score of our proposed model is 0.93, 0.84 and 0.88 respectively. Comparison with state-of-the-art architectures demonstrated better performance of SarNet by a significant margin.

## 1 Introduction

Social media has the potential to influence the opinion of the masses. People from all around the world interact and share their perspectives. This has facilitated the rapid exchange of ideas. Unfortunately, these social media platforms are being used to disseminate hate speech. Hate speech is described as aggressive or threatening speech that shows prejudice based on ethnicity, religion, sexual orientation or other factors. This raises the need to deploy robust and efficient classifiers to regulate online content.

Citizens on social media regularly use sarcasm to convey their emotions in conversation. Sarcasm is an effective means of expressing thoughts indirectly that is not easy to notice. It is described as an incidental technique of expressing a viewpoint in which the written word does not reflect the intended meaning. Most of the hate speech detection algorithms (1; 2) face difficulty in distinguishing sarcastic statements. This is because sarcasm reverses the polarity of a seemingly positive or negative phrase. This attribute of sarcasm leads hate detection classifiers to falsely flag sentences as hate. Thus, it is crucial to reduce the false positives by considering the sarcastic context of a sentence while predicting hate.

The labelled datasets suffer from annotators' bias and/or the unavailability of datasets annotated as both hate and sarcasm (5). The above limitations of datasets make it very difficult for deep learning models to predict the actual hate content of a tweet with high precision. Our proposed SarNet model overcomes these problems by employing a semi-supervised learning approach to predict the degree of hate in a sentence that we define as true-hate.

Researchers have presented multiple approaches for tackling hate speech in the past decade. Kapil and Ekbal (14) proposed a deep neural network-based multi-task learning approach for hate speech detection. Corazza *et al*. (4) proposed a neural network classifier for multilingual online hate speech detection in three languages and investigated the impact of each feature on the respective outcomes. While it is critical to identify hate on social media promptly, it is equally important to avoid false positives. Arango *et al*. (5) point to methodological flaws as well as significant dataset biases while tackling hate speech. (5) stated that many state-of-the-art performance claims had become vastly

overstated and suggested a more realistic view of the state-of-the-art methodologies. Meriem *et al.* (6) presented a fuzzy sarcasm detection method that uses social knowledge and multiplies it by a degree of relevance. Kumar *et al.* (7) presented a multi-head attention-based BiLSTM network for detecting sarcastic remarks in a corpus. The aforementioned works do not consider the sarcastic and hate context in the same tweet while prediction.

Most of the methods (8; 9; 10) fail to consider the sarcastic context of the sentence while detecting hate speech, which leads to misclassification of a sarcastic sentence as hate. Thus, we propose the SarNet model to tackle the misclassification problem in hate speech detection by considering the sarcastic context of the sentence.

This paper presents SarNet, a two-fold learning method wherein we employ the concept of multiple games based on game theory (Prisoners' Dilemma) and Nash equilibrium. SarNet detects true-hate by considering the sarcastic context of the sentence and gives a pooled output. Hate and sarcasm are a perspective-dependent sentiment. Our proposed SarNet model as shown in Figure 10 analyzes the contextual information of a sentence. The major contributions of this work are summarized as follows:

- Introduce a novel model, SarNet, by integrating the ANN, LSTM network for calculating probabilities to Nash Equalizer which plays games to identify the label of the sentence, using a quasi-ternary labeling technique .
- Perform comparative analysis of SarNet to establish its supremacy.
- Investigate the impact of different hyper-parameters' values on the efficacy of SarNet, and the effectiveness of the Nash Equalizer.

The paper is structured as follows: Section II is a review of existing works. Section III describes SarNet at an intuitive level and presents the experimental and hyperparameter settings and results, compares them with the current state-of-the-art and discusses the model. Section IV draws a conclusion to our research work and Section V presents the limitations of our work, giving future directions.

## 2 RELATED WORK

This section summarizes existing research in hate speech detection and sarcasm detection in social media. The current strategy is split into four; namely, machine learning based approaches, deep learning based approaches, sarcasm detection and hate detection. We discuss all the sub-categories in the following subsections.

### 2.1 MACHINE LEARNING BASED APPROACHES

Dave and Desai (11), attempt to identify various supervised classification techniques like SVM, Naïve Bayes using basic Bag-of-words features with TF-IDF as the feature's frequency measure. The researchers studied the outcomes of these classification approaches using textual data accessible in multiple languages on review sites, social networking sites, and microblogging sites, as well as the data set construction and feature selection processes. A pilot experiment is carried out by the researchers to detect sarcastic statements in "Hindi."

Koushik *et al.* (12) propose a machine learning technique for automatically detecting hate tweets utilising a bag of words and the TFIDF approach on an existing Twitter data set. The researchers utilised a logistic regression classifier for binary categorisation of the tweets. The researchers obtained 94.11 per cent accuracy using a bag of words feature, and 94.62 per cent accuracy using the TFIDF feature.

### 2.2 DEEP LEARNING BASED APPROACHES

Patro *et al.* (13) present a deep learning framework for detecting sarcasm targets in pre-defined sarcastic messages by conducting empirical research of socio-linguistic variables and identify those that are statistically significant in predicting sarcastic targets. Finally, they provide a deep learning LSTM based architecture enhanced with sociolinguistic features on a dataset released by Joshi et al.

Kapil and Ekbal (14) propose a deep multi-task learning framework to harness beneficial information from numerous related categorization tasks to improve the performance of the individual task. The multi-task model is based on a shared-private architecture in which shared and private layers are assigned to capture shared and task-specific characteristics. Experiments on five benchmark binary datasets suggest multi-task model performs better than the single-tasking framework and indicate that separate data sets classified in different sub-classes do indeed aid each other in categorization.

An automated system is proposed by Roy *et al.* (15) using the Deep Convolutional Neural Network (DCNN) which utilises the tweet text in English language with GloVe embedding vector to capture the tweets' semantics with the help of convolution operation. Besides this the researchers also experimented on state of the art machine learning classifiers. The researchers claim that they achieved the precision, recall and F1-score values as 0.97, 0.88, and 0.92 respectively on the DCNN network.

Plaza-del-Arco *et al.* (16) address the task of identifying Spanish hate speech on social media. They specifically compare Deep Learning methods to recently pre-trained language models based on Transfer Learning as well as classic machine learning models. The researchers claim that their findings produced with the TL models surpass the other ML models. They evaluate the performance of two multilingual pre-trained models (mBERT and XLM) that are currently accessible with a monolingual model (BETO) that has been particularly trained on Spanish. The results demonstrate that the monolingual pre-trained LM (BETO) outperformed mBERT and XLM.

## 2.3    SARCASM DETECTION

Researchers are continuously working on detecting sarcasm in sentences.(11), developed a method to identify sarcasm using various supervised classification techniques like SVM, Näıve Bayes using simple BoW features and TF-IDF as the feature's measure of frequency. The researchers examined these classification techniques' results on textual data available in various languages on various websites along with the analysis of dataset generation and feature extraction process.

Researchers have used deep learning-based classifiers for the detection of sarcasm. Patro *et al.* (13) provide a deep learning framework for recognising sarcasm targets in predefined sarcastic messages by conducting an empirical study on socio-linguistic characteristics that are statistically important in predicting sarcastic targets. Husain *et al.* (17) present a methodology based on the hypothesis that tweets with negative sentiment and sarcastic content are more likely to contain offensive content; thus, fine-tuning the classification model using a large corpus of offensive language which aids the model's learning process for detecting sentiment and sarcastic content, and investigate transfer learning in the contexts of foul language, sarcasm detection, and sentiment analysis. Using the mathematical formalisms of quantum theory and fuzzy logic, Zhang *et al.* (18) suggest a complex-valued fuzzy network. Verma *et al.* (19) focus on new research on sarcasm detection, including several strategies and problems based on deep hybrid learning. Nayel *et al.* (20) propose an SVM-based classifier for sarcasm detection in Arabic language. The researchers perform a model comparison with other state-of-the-art techniques.

## 2.4    HATE DETECTION

The NLP community has constantly been working on the problem of hate speech classification. Pitsilis *et al.* (21) describe a hate speech detection strategy that integrates several user-related characteristics, such as the users' tendency for racism or sexism, with an ensemble of RNN classifiers. Pachalides *et al.* (22) proposed MANDOLA. This big-data processing system uses big-data methodologies to monitor, detect, display, and report the expansion and penetration of online hate-related discourse. Teh *et al.* (1) looked into how profanity was used on Twitter by people of different groups to mark how successful it was in detecting hate speech. To rationalize the differences in profanity usage across the three nations, statistical hypothesis tests are conducted, and a Bayes theorem-based probability estimation process is developed to quantify the efficacy of blasphemy-based detection systems for hate speech. Senarath *et al.* (23) claim that semantic information can aid machine learning algorithms in augmenting the word senses that are represented in context to social media posts and present a new empirical study of various semantic variables such as word embedding representation for distributional semantics, corpus-based semantics and declarative knowledge patterns for hate speech categorization tasks on social media posts. Mozafari *et al.* (24) presented a transfer learning strategy based on BERT by evaluating the potential of BERT to capture hateful context

inside social media content using various fine-tuning approaches. Caselli *et al.* (25) proposed Hate-BERT, a re-trained BERT model trained on a publicly available Reddit comments data set for hate language detection in English.

# 3 METHODOLOGY

SarNet is proposed for true-hate detection which includes data pre-processing, and an explanation of our proposed two-fold deep learning based method.

## 3.1 DATA PRE-PROCESSING

Useless information is filtered by pre-processing the datasets. In the pre-processing, SarNet performs the following steps: The tweets were converted into lower case and the words were split for removal of stop words using the NLTK library. Using regex white spaces, URLs, hashtags, twitter handles, emoticons, dots, punctuation marks and non-ASCII characters etc. were removed. The words were then stemmed using SnowBall Stemmer.

## 3.2 PROPOSED SARNET MODEL

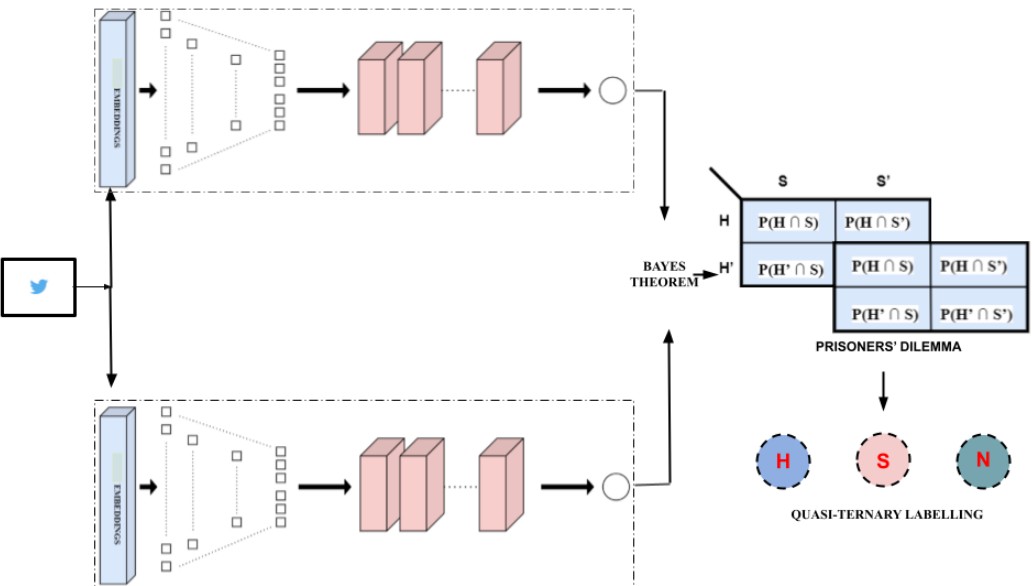

Figure 1: Detailed Architecture of SarNet

Figure 1 shows the SarNet model. SarNet is a two fold deep learning model. It is used for quasi ternary labeling of tweets as hate, sarcasm or neither. The SarNet model is built using two parallel Pyramid Networks. They are trained on a hate speech dataset and a sarcasm dataset respectively. These parallel pipelines converge into the Nash Equalizer.

The Pyramid Network of the model is built using an input layer followed by an embedding layer which passes the data to ANN layers which consist of regular dropouts. This helps in faster feature extraction. These layers are integrated with stacked LSTM layers followed by a sigmoid activation which helps in predicting the probability of a given tweet. These probabilities are then fed into the Nash Equalizer for the final labelling. The model is discussed at an intuitive level in the following subsections.

The pre-processed tweet is passed to the input layer based on the token generated by indexing the dictionary values. As a result, the input layer transforms the incoming text into a vector.

The embedding layer semantically correlates the relevant terms with equivalent vector representations. We use pre-trained (GloVe) (26) embedding for word representation.

### 3.2.1 Pyramid Network

The Pyramid Network exploits successive ANN layers to extract features from the pre-processed input tweet vectors. We do not use CNN layers as the words in a tweet do not have spatial correlation. There are multiple successive layers of ANN and dropout for downsizing the initial number of features extracted by the ANN layer. The dropout between successive layers is by a factor of either 0.2 or 0.4 (refer Section 3.5 Hyperparameter Settings) . Finally six most relevant features (calculated empirically to produce best results) per tweet are passed onto the LSTM layers for identifying sequential correlation between them. The ANN layers help in faster and accurate feature extraction which helps in reducing the training time significantly when compared to other feature extractors like RNN. Equation 1 represents the nth feature sequence , $z_n$, which is a summation of the dot product between the matrix of weights $w^\mathrm{T}$ and input tweet row vector $x$ and the bias $b$. $f(.)$ is ReLU (non linear activation function). $d$ gives the factor of dropout which is either 0.2 or 0.4.

$$z_n = d * f(w^T.x + b) \; ; \;\; d = \{0.2, 0.4\} \tag{1}$$

BiLSTM employs a memory cell that is made up of an input gate, a forget gate, an output gate, and a node that is linked to itself. The memory cell in a certain layer utilises the previous layer's hidden state at the current time and the current layer's hidden state from the prior time. The forget gate determines which information in the cell state should be discarded, while the input gate and $tanh$ layer determine which information is stored in the cell state. The best features extracted from the ANN layers are passed onto multiple successive BiLSTM cells to map the selected features sequentially. BiLSTMs enable additional training by traversing the input data in the forward and the backward directions thus reinforcing the contextual information extracted. The extracted information is passed through a sigmoid activation layer for calculating the probabilities of prediction which are then passed onto the Nash Equalizer. BiLSTM alone is not self sufficient to gather the context of hate and sarcasm of a given tweet. To overcome this ambiguity a payoff matrix using Nash Equilibrium is constructed which along with probabilistic terms quantify the tweet as hate or sarcasm.

### 3.2.2 Nash Equalizer

As discussed above, it is crucial to consider the context of the sentence, that is its intent and estimate a degree of hate or sarcasm of that sentence before classifying it. Many a times it is difficult to determine whether a sentence is hate or not as its meaning cannot be understood at its face value. This is especially true when sarcasm is used, as the true intent of the sentence is hidden behind words. At its face value the sentence might seem to spur hate but in reality the intention behind might be the opposite. Thus to get an estimate of the hatefulness of a sentence taking into account the sarcastic part of the sentence we propose a parameter true-hate.This proposed parameter gives a measure of the hate content of a given tweet excluding the sarcastic part. It will greatly help in assessing the degree of hate content in a tweet and also in reducing false flagging of not hate tweets. Thus we propose our Nash Equalizer which predicts the degree of hate of a tweet.

The pyramid network being trained on the hate speech dataset learns the hate content features of a sentence and the pyramid network being trained on the sarcasm dataset learns the sarcasm content features of a sentence. Thus, prior to testing, we acquire two pyramid networks that have learned two distinct semantics of a sentence, namely hate and sarcasm. Following the initial training of both pyramid networks, the test split ( refer Section 3.3 Datasets of the paper for detailed explanation of the test split) of the hate speech dataset is simultaneously passed through both pipelines. Let's assume a tweet $t$ labelled as hate from the hate speech dataset's test split. It is simultaneously passed through both pipelines of the pyramid network. The hate pyramid network will calculate the hate probability of tweet $t$. The sarcasm pyramid network returns the sarcasm probability of the given tweet $t$. As a result, for the tweet $t$, we will get the probabilities of hate and sarcasm.

Since both the probabilities are obtained from two different pipelines, they are independent from each other. Bayes Theorem of conditional probability is now used to calculate four probabilities which are as follows:

- Probability of a tweet being sarcastic given hate, ($P_{S|H}$ ).
- Probability of a tweet being not sarcastic given hate, ($P_{S'|H}$).

|          | Sarcasm     | Not Sarcasm     |
|----------|-------------|-----------------|
| Hate     | $P_{S|H}$   | $P_{S'|H}$      |
| Not Hate | $P_{S|H'}$  | $P_{S'|H'}$     |

Table 1: Payoff Matrix

- Probability of a tweet being sarcastic given not hate, ($P_{S|H'}$).
- Probability of a tweet being not sarcastic given not hate, ($P_{S'|H'}$).

The following equation is used to obtain four cases of probabilistic relationships.

$$P_{S|H} = \frac{P(H \cap S)}{P(H)} \tag{2}$$

Finally the Nash Equalizer constructs a Prisoners' dilemma considering sarcasm and hate as the two prisoners as shown in Table 1. The pooled output of the table constructed using the dilemma is used to calculate the label for the final tweet as follows:

- Case I: The maximum probability of a tweet comes out to be not sarcastic given not hate. This case is clearly indicative of the fact that the given tweet is neither hate nor sarcastic. Thus this case of tweet is labelled as neither hate nor sarcastic.

- Case II: The maximum probability of a tweet comes out to be not sarcastic given hate. This case is clearly indicative of the fact that the tweet is hate and not sarcastic. Thus this case of tweet is labelled as hate.

- Case III: The maximum probability of a tweet comes out to be sarcastic given not hate. This case is clearly indicative of the fact that the tweet is not hate but is only sarcastic. Thus this case of tweet is labelled as sarcastic.

- Case IV: The maximum probability of a tweet comes out to be sarcastic given hate. This case is clearly indicative of the fact that the tweet is hate as well as sarcastic. For this case we consider true hate factor instead of hate.

As mentioned above for cases I, II and III we do not consider the true hate of any tweet. But if the tweet belongs to category IV, that means that the given tweet contains hate content as well as sarcasm content. We then proceed to evaluate the true hate parameter of the tweet. This is the case wherein most classifiers fail to flag the nature of the tweet correctly. Thus to tackle this we propose the true hate parameter. The true hate factor considers effective hate by excluding the sarcastic component of the given tweet. These include cases where the sentence looks like hate but is sarcastic. For calculating the true hate factor we start with simple hypothesis and move towards complex relations. A simple relationship calculates true hate by subtracting the probability of sarcasm from hate and if the probability of hate is greater than sarcasm then the given tweet is flagged as hate and vice-versa. The hypothesis used to calculate true hate is given as follows:

$$P(H_{True}) = P(H) - \frac{[P(S)]^n}{m} \tag{3}$$

Here $n, m \ \epsilon \ N$ This hypothesis is used to evaluate the probability of the true hate ($H_{True}$) parameter of a tweet in case where the tweet is sarcastic given hate. As seen in the equation for such a case we revert back to the original hate and sarcasm probabilities of the tweet and from those we obtain the true hate of such a tweet. If we obtain the probability of true hate to be greater than 0.5, we classify the tweet as hate otherwise we classify the tweet as sarcastic.

## 3.3 DATASETS

We evaluate our proposed SarNet model on 2 benchmark datasets.

**Hate Speech Dataset:**

Davidson *et al.* (27) utilised Twitter API and collected 84.4M tweets from 33.458K users. The CrowdFlower crowdsourcing platform annotated the randomly sampled 25k tweets. Since the dataset was unbalanced we had to deploy an oversampling technique to balance the classes. We then binarised the classes as hate and not hate.

Basile *et al.* (29) outlined the structure of the SemEval-'19 Task 5 identifies hate speech against hate speech against immigrants and women in Spanish and English. The data set is composed of 6,600 spanish and 13,000 english tweets. We utilised the english corpus for our hate speech training purposes.

For the task of training one pipeline of the pyramid network on hate speech we binarised both the datasets (27) and (29) and merged them into one dataset. This was done to increase the size of the dataset. The total number of hate tweets were 37,521 and the total number of not hate tweets were 45,045. This new combined hate speech dataset was then split into train, validation and test. The train, validation and test split ratio is 70, 20 and 10 percent respectively. Fig 2(a) is a bar graph which shows the number of hate and not hate tweets.

We tested our true hate detection task on another hate speech dataset (30) The dataset was created using Twitter data. The text is divided into three categories: hate speech, offensive language, and neither. For our task we binarised the class labels as hate and not hate by merging the hate speech and offensive language class labels into one class label; i.e. hate. The dataset consisted of 2,242 hate tweets and 29,720 not hate tweets. Due to such imbalance of the dataset we applied SMOTE to upsample the hate tweets to 29,720 hate tweets. Figure 2(c) is a bar graph which shows the number of hate and not hate tweets after SMOTE has been applied.

**Sarcasm Dataset:**

Arora *et al.* (28) proposed a binary labelled sarcasm detection dataset in english. The sarcastic statements were extracted from 2 news websites: the Onion, which publishes satirical renditions of critical events, and all of the headlines from the News in Brief and News in Photos categories. HuffPost provided the non-sarcastic news headlines. The advantage of using this dataset was its high quality labels with little noise, formal usage of english language and absence of spelling mistakes.

For the task of training one pipeline of the pyramid network on sarcasm detection we used (28) dataset. It consisted of 13,624 sarcastic tweets and 14,985 not sarcasm tweets. The train, validation split for this dataset is 80 and 20 percent respectively. Fig 2(b) is a bar graph which shows the number of sarcasm and not sarcasm tweets.

## 3.4 Experimental Settings

SarNet is implemented using Python 3. All the experiments were performed on Google Collaboratory[1]. The GPU is Tesla K80 with a memory of 12 GB. Tensorflow[2] , API was used.

## 3.5 Hyperparameter Settings

The hate dataset is split into training, validation and testing 70, 20 and 10 per cent respectively while the sarcastic dataset is split into train and valid of 80 and 20 respectively. The hate pipeline of the Pyramid Net trains and fine-tunes on the train and validation splits of the hate dataset and the sarcastic pipeline of the Pyramid Net trains and fine-tunes on the train and validation split of the Sarcastic dataset. Both the pipelines are used to calculate the probabilities on the test split of the hate dataset. Furthermore the total number of ANN layers in the Pyramid Net are 7 with neurons decreasing from 100, 80, 60, 40, 20, 10 to 6. Dropout layers are present with a factor of 0.2 from ANN layers of 100 to 10 and a factor of 0.4 from 10 to 6. The output vector of the ANN layer of 6 neurons is transposed into a 2- dimensional matrix using a repeatvector layer. The 7 BiLSTM layers contains 100 cells each. ANN layers use ReLU activation function. Adam optimizer and the bininary cross-entropy loss function are used. Total trainable parameters of the Pyramid Network are 1,697,177.

---

[1]https://colab.research.google.com
[2]https://www.tensorflow.org

## 3.6 COMPARATIVE ANALYSIS

In this section we give a detailed comparative analysis of our proposed SarNet model with 14 baseline models and 5 state-of-the-art models.As a result, we recreate the comparison techniques from scratch using the instructions provided in those publications. Our proposed model SarNet achieved the best results. It achieved a precision, recall and an F1 score of 0.93, 0.84, 0.88 respectively.

**Comparison with Baseline Models:**

SarNet was compared with 14 baseline models. In this comparative analysis various combination of feature extraction techniques followed by classifiers were tested for the hate speech detection task on new combined hate speech dataset. The formation of the new combined dataset is described above in subsection 3.3. The following techniques were used for feature extraction:

- TF-IDF- The relevance of a term is represented by (Term Frequency-Inverse Document Frequency). It is the result of calculating the dot product of TF and IDF.

- GloVe- GloVe (26) is an abbreviation for Global Vectors for Word Representation. It is an unsupervised learning approach that seeks to construct word embeddings from a given corpus by aggregating global word co-occurrence matrices. The primary idea behind the GloVe word embedding is to use statistics to determine the link between words.

- Small BERT- BERT (32) base consists of 12 layers, 768 hidden size and 12 attention heads.

- BERT- BERT large (33) consists of 24 layers, 1024 hidden size and 16 attention heads.

- ALBERT- ALBERT (35) is a "Lite" version of BERT with much less parameters. ALBERT (like BERT) uses a deep neural network using the Transformer architecture to calculate dense vector representations for natural language.

- ELECTRA- ELECTRA (34) is a BERT-like model that has been pre-trained as a discriminator in a generative adversarial network setup (GAN). ELECTRA (like BERT) uses a deep neural network using the Transformer architecture to calculate dense vector representations for natural language.

The baseline classifiers used for the hate speech detection task are SVM, XGB, MLP, CNN and Bi-LSTM. The combination of feature extraction techniques and classifiers deployed can be seen in the bar graphs given in the appendix section. The best performance in terms of precision was obtained by BERT+CNN combination in hate speech detection where BERT was used for feature extraction and CNN was used for classification. It achieved a precision of 0.78. In terms of recall Small BERT+CNN achieved a best recall of 0.81. The best F1 score was achieved for ELECTRA+MLP and BERT+CNN. The F1 score achieved was 0.76.

**Comparison with State-of-the-art Models:**

5 state-of-the-art models were trained on the new combined dataset for hate speech detection. They are as follows:

- Siino et al. (8) presented a deep learning model based on a 1-D CNN. Their model achieved a precision, recall and an F1 score of 0.81, 0.49 and 0.79 respectively when trained and tested on the new combined hate speech dataset.

- Pereira-Kohatsu et al. (37) proposed HaterNet wherein they deployed multiple deep learning based models. We trained their best performing model which used Tf-Idf for feature extraction followed by a combination of MLP + neural networks for classification. When trained and tested on our new combined hate speech dataset their model achieved a precision, recall and F1 score of 0.66, 0.49 and 0.64 respectively.

- Roy et al. (15) proposed a deep CNN based model using GLoVe embeddings. Their model achieved a precision, recall and F1 of 0.61, 0.42 and 0.48 respectively.

- Ding et al. (38) proposed a stacked BiGRU model based on a capsule network architecture. They used fastText to transform tweets into vector representations. Their model achieved a precision, recall and F1 score of 0.67, 0.64 and 0.62 respectively when trained and tested on the new combined hate speech dataset.

- Khan et al. (36) proposed HCovBi-Caps a convolutional, BiGRU, and capsule network-based deep learning model using GloVe embeddings. Their model when trained and tested on our new combined hate speech dataset achieved a precision, recall and F1 score of 0.84, 0.75 and 0.76 respectively.

## 4 CONCLUSION

In this research we proposed SarNet, a two fold deep learning model to predict the true hate of a given tweet. Our main goal was to extract the degree of hate and sarcasm from a tweet to get a more realistic comprehension of any given tweet. For this task we proposed the pyramid network to extract the probabilities of hate and sarcasm. Two parallel pipelines of the pyramid network were trained on sarcasm and hate speech datasets to get their probability. Using Bayes' Theorem we made four cases of probabilities of hate and sarcasm for any given tweet. Then our proposed Nash Equalizer used the concept of game theory (Prisoner's Dilemma) to construct a payoff matrix to give one of the four cases of degree of hate and sarcasm. From here we calculated the true hate parameter for a particular case ($P(H|S)$) obtained from the Nash Equalizer. The rest of the three cases were directly classified. In the end tweets were classified as hate, sarcastic or neither. We used one Kaggle sarcasm dataset for learning the sarcasm features of the tweets. We trained and tested our hate speech task on the new combined dataset which was formed by merging the T.Davidson dataset and the SemEval-2019 Task 5 english hate speech dataset. Our model achieved a precision, recall and F1 score of 0.93, 0.84 and 0.88 respectively. We trained and implemented 14 baseline classifiers and 5 state-of-the-art architectures on the new combined dataset for comparison with our model. Our model outperformed all other classifiers. We studied the impact of various hyperparameters on the efficacy of our proposed SarNet model.

## 5 LIMITATIONS

The main drawback is the non availability of robust datasets. Datasets suffer from multiple problems such as class imbalance, annotator bias, difference in the definition of hate speech and varying forms and intentions of expressions. The availability of robust datasets would increase the validity of our model's prediction. We have researched only on english. Research work can further be extended to other languages, as well as considering other parameters, alongside sarcasm like sexism etc.

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

## 6 APPENDIX

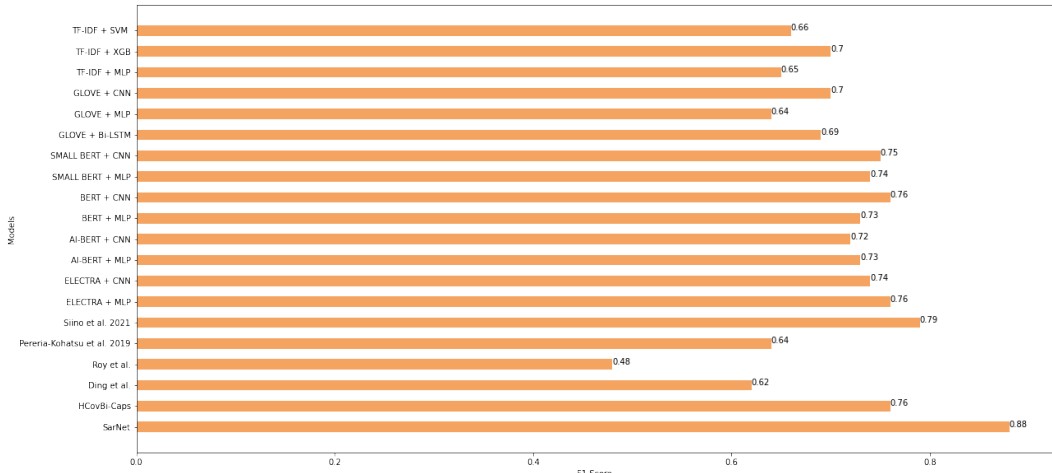

Figure 2: F1 Score of SarNet vs state-of-the-art on T.Davidson + SemEval Dataset

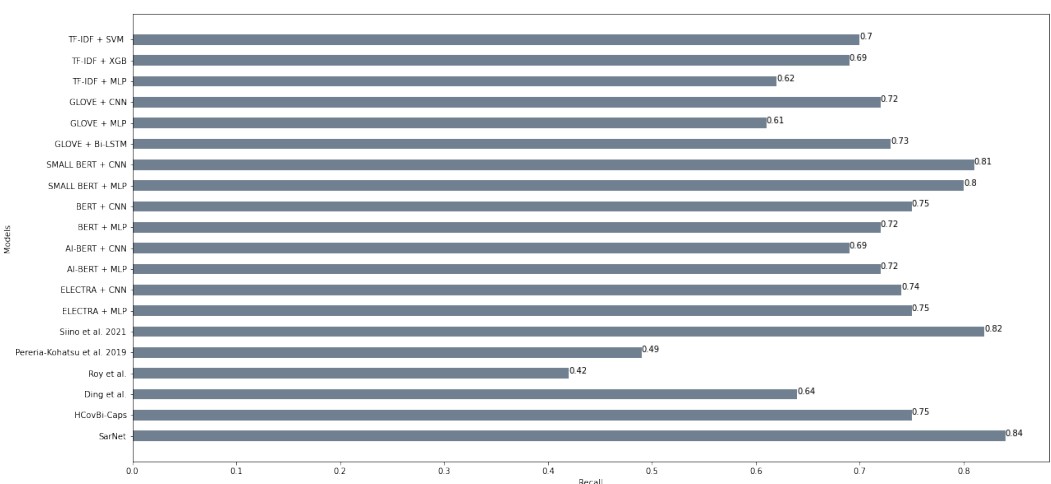

Figure 3: Recall Score of SarNet vs state-of-the-art on T.Davidson + SemEval Dataset

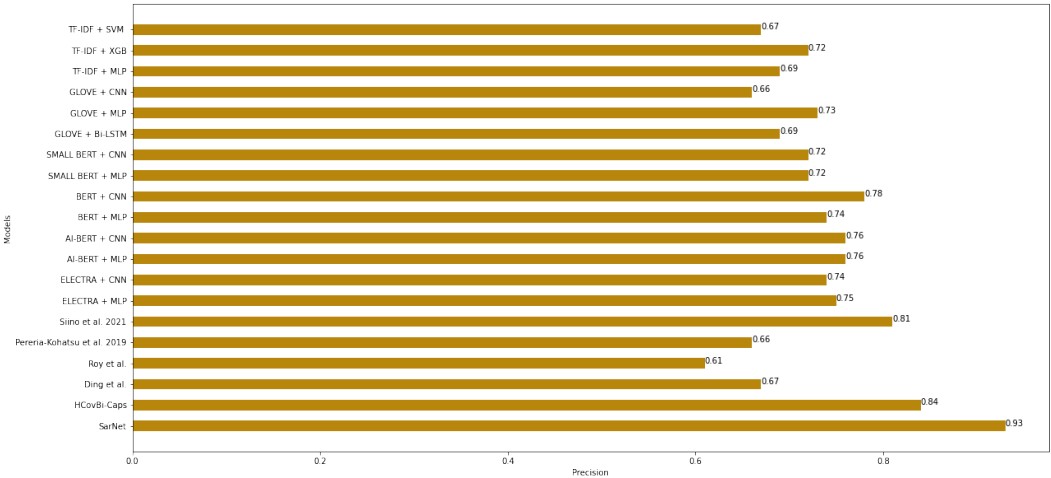

Figure 4: Precision Score of SarNet vs state-of-the-art on T.Davidson + SemEval Dataset

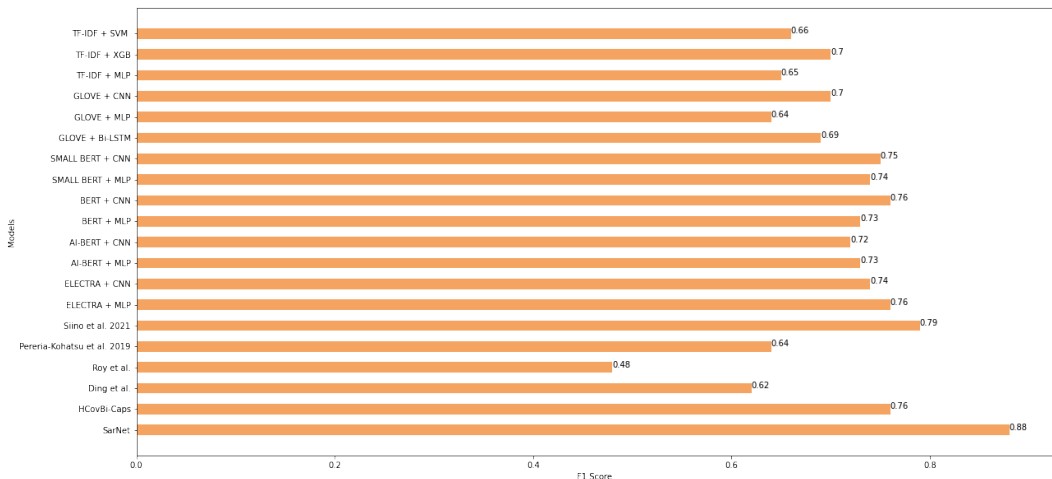

Figure 5: F1 Score of SarNet vs state-of-the-art on T.Davidson + SemEval Dataset

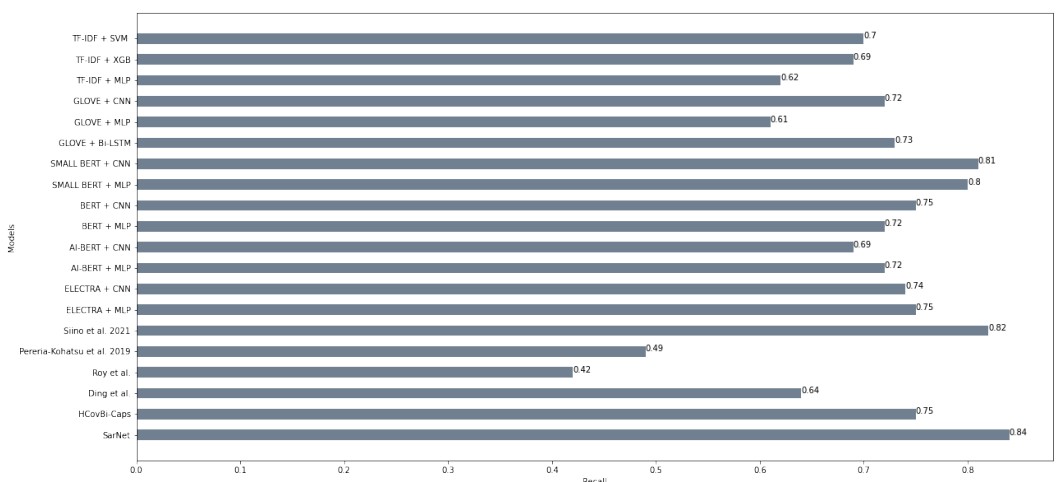

Figure 6: Recall Score of SarNet vs state-of-the-art on T.Davidson + SemEval Dataset

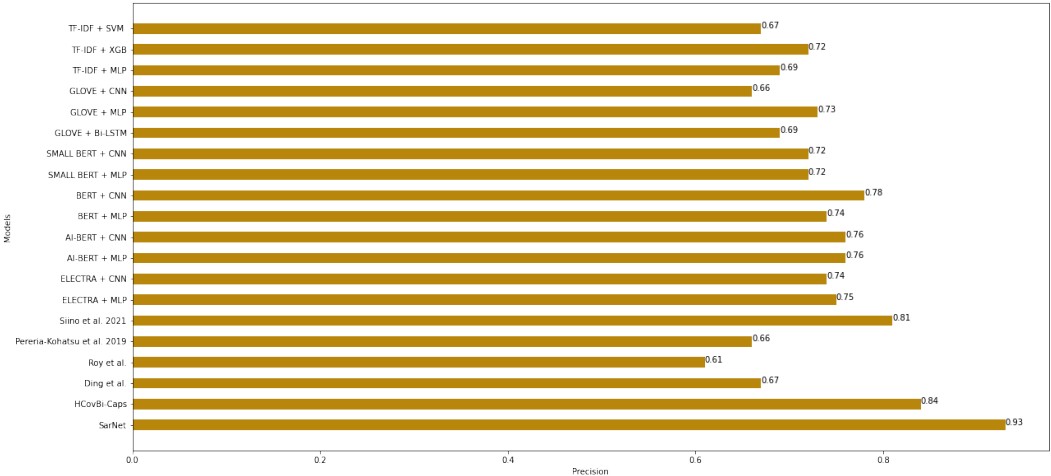

Figure 7: Precision Score of SarNet vs state-of-the-art on T.Davidson + SemEval Dataset

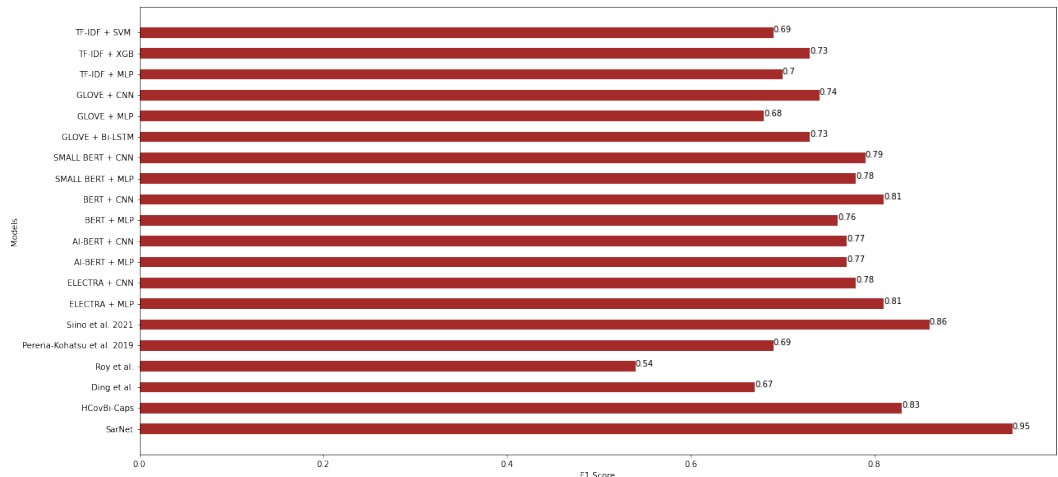

Figure 8: F1 Score of SarNet vs state-of-the-art on Kaggle Dataset

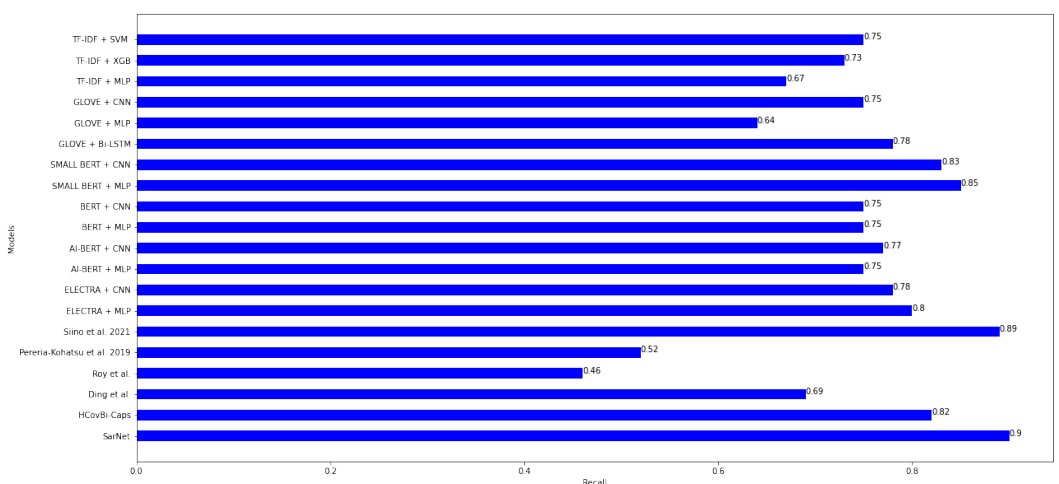

Figure 9: Recall Score of SarNet vs state-of-the-art on Kaggle Dataset

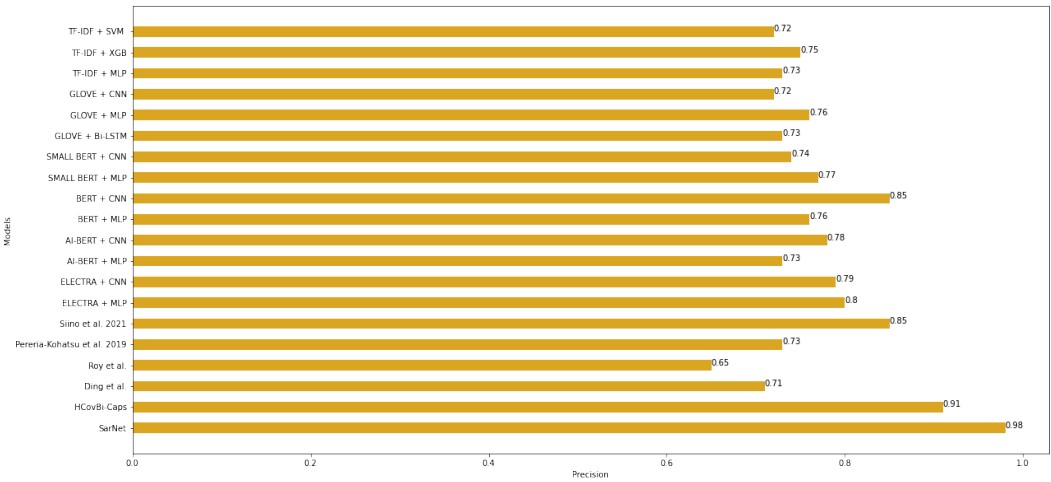

Figure 10: Precision Score of SarNet vs state-of-the-art on Kaggle Dataset

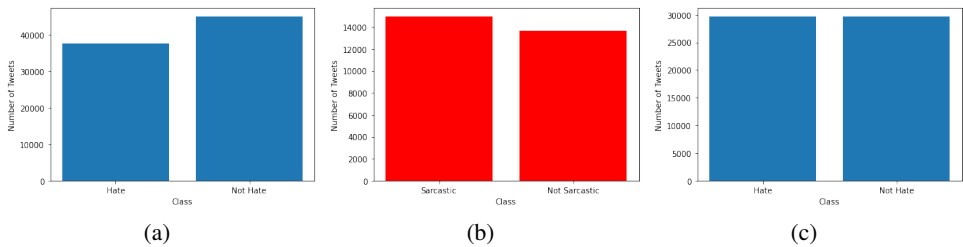

Figure 11: Dataset for (a) Hate Tweets (b) Sarcastic Tweets (c) Kaggle Hate after SMOTE

