# OpenReview forum: "SARNET: SARCASM VS TRUE-HATE DETECTION NETWORK"
_ICLR.cc/2023/Conference — Submitted to ICLR 2023_

### Official Review · Reviewer_jETC · 2022-10-24

**Confidence:** 3
**Correctness:** 3
**Technical Novelty And Significance:** 2
**Empirical Novelty And Significance:** 1
**Recommendation:** 5

**Clarity, Quality, Novelty And Reproducibility:**

In terms of clarity, the article needs to be further refined in both the logic part and the expression part.
In terms of novelty, SarNet is not the first method to tackle hate speech detection tasks.
In terms of reproducibility, looking at this article from METHODOLOGY, it is not difficult to reproduce this work.
In terms of quality, this paper proposes a two-fold deep learning-based model, which can achieve performance improvements, but the logic and expression of the article need to improve.

**Strength And Weaknesses:**

Strengths
1. The author constructs a prisoner's dilemma via a Nash equalizer, treating sarcasm and hate as two prisoners, and finally uses the output of the dilemma to calculate the label of the final tweet.
2. Improved performance compared to baseline methods and state-of-the-art.

Weaknesses:
1. The content of the article needs to be further refined.
- Introduction mentions "Our proposed SarNet model as shown in Figure 10 analyzes the contextual information of a sentence.", but what Figure 10 expresses is "Precision Score of SarNet vs state-of-the-art on Kaggle Dataset", which does not fit the context.
- There is an extra space in "94.11 per cent" and "94.62 per cent" in Sec 2.1.
- The semantics of the two sentences in "SarNet is proposed for true-hate detection which includes data pre-processing, and an explanation of our proposed two-fold deep learning based method." mentioned in Methodology are discontinuous.
- The picture of Figure 1 is so blurry that I can't even make out the text in the blue box on the left.
- In the experimental part, the authors did not analyze the experimental results.
- There is no table or picture in the experimental part of the text, and the related work part is too long.

**Summary Of The Paper:**

In the paper, the authors propose SarNet, a method to tackle the hate speech detection task. The authors argue that previous methods tend to judge satirical speech as hate speech, resulting in false positives. Their main goal was to extract the degree of hate and sarcasm from a tweet to get a more realistic comprehension of any given tweet. They propose the pyramid network to extract the probabilities of hate and sarcasm.

**Summary Of The Review:**

This paper can achieve a certain performance improvement on hate speech detection task, but the logic and expression of the article have many areas worth improving.

---

### Official Review · Reviewer_cP5A · 2022-10-25

**Confidence:** 3
**Correctness:** 3
**Technical Novelty And Significance:** 2
**Empirical Novelty And Significance:** 2
**Recommendation:** 5

**Clarity, Quality, Novelty And Reproducibility:**

Generally, the manuscript starts off well with the sections on introduction and related work. In terms of quality, this paper presents a two-fold deep learning-based model, which can reach performance improvements, but the logic and expression of the article need to improve. In terms of clearness, the paper must be further refined in both logic and expression. Regarding reproducibility, glancing at this paper from a methodology perspective, it will be challenging to replicate this work. About novelty, SarNet is not the preferred method to tackle hate speech detection tasks.

**Details Of Ethics Concerns:**

No comments

**Strength And Weaknesses:**

Still, the ANN layers in the pyramid network pull aesthetic elements from the processed input tweet vectors. What would the model do with input texts that are often short primary samples with confusing words like idioms, onomatopoeias, homophones, phonemes, synonyms, acronyms, anaphora, and polysemy?

ANN can solve various problems, including virtually any problem reduceable to functions, yet, sharing is challenging after training an ANN; overfitting and convergence cannot be guaranteed. So, how will the SarNeis model handle these challenges? The addition of
nash equilibrium to quantify probabilistic representations of the tweet as hate or sarcasm is excellent, but what type of stylometric features did ANN extract and improve with the BiLSTM layer?


**Summary Of The Paper:**

This paper presents two deep learning methods, SarNeis, that address the nuisance of hate speech detection in a given text or sentence and incorrectly flag a sarcastic tweet. First, the author employed ANN-BiLSTM, a pyramid network, to compute a sentence's hate and sarcastic probabilities. Secondly, the author used the nash equalizer from the game theory concept and prisoner's dilemma. In summary, SarNet integrates the ANN and LSTM network for calculating probabilities to Nash Equalizer, which plays games to identify the label of the sentence, using a quasi-ternary labeling process. The proposed method treats hate and sarcasm as two prisoners.

**Summary Of The Review:**

An interesting method is presented, focusing on using ANN-BiLSTM as a pyramid network to calculate a sentence's hate and sarcastic probabilities. The presented results show competitive performance with the SOTA method during training.  The paper is missing an error analysis section to investigate the registers of misclassified texts critically. Please provide examples of correctly/misclassified text with critical reflection.

---

### Official Review · Reviewer_xKDB · 2022-10-26

**Confidence:** 3
**Correctness:** 3
**Technical Novelty And Significance:** 3
**Empirical Novelty And Significance:** 2
**Recommendation:** 5

**Clarity, Quality, Novelty And Reproducibility:**

- it seems novel and original. overall it was clear to describe what was done.

**Strength And Weaknesses:**

strength:
- sarcasm in hate speech seems important, and this paper tackles that problem.
- the proposed model outperforms baselines and state-of-the-art models for hate speech detection

weaknesses:
- I am not familiar with game theory (Prisoners’ Dilemma) and Nash equilibrium, etc. presented in this paper, and had difficulty to truly understand the proposed model.
- Although the experiment results suggest that the detection performance of the proposed model is superior to other models, more analysis/discussion (especially regarding sarcasm) will be helpful to show whether the proposed model works in an intended way or not.
- the study about the impact of different hyperparameter settings? Where can I find them?

**Summary Of The Paper:**

This paper proposes a model for detecting true hate (vs. sarcasm) speech in texts, i.e., hate speech detection taking into account sarcasm.  The proposed model uses game theory (Prisoners’ Dilemma) and Nash equilibrium. The experiments show that the proposed model outperforms baselines and state-of-the-art models.

**Summary Of The Review:**

The paper tackles the problem of sarcasm in hate speech, and the proposed model outperforms other baselines and state-of-the-art. However, some analysis or discussion to show the efficacy of the model will be helpful.

---

### Official Review · Reviewer_ioZ4 · 2022-11-02

**Confidence:** 5
**Clarity, Quality, Novelty And Reproducibility:** Clarity and quality of exposition is …
**Correctness:** 3
**Technical Novelty And Significance:** 2
**Empirical Novelty And Significance:** 2
**Recommendation:** 3

**Strength And Weaknesses:**

Strengths:

- Good idea to try to separate out hate from sarcasm via separate network.

Weaknesses:

- Paper is clearly not in a finalized state, it needs a thorough efforts in polishing. From Figures to how literature is compared in bullet point fashion, without trying to synthesize it.
- Ideas also are too simple. Of course simple ideas can also work and those would definitely be valuable, but then authors need a bullet proof experiments. To really show that ideas work.
- Experiments are basically non-existent. Authors should note that main paper needs to have the main story supported by the experiments, anything that is in appendices are not necessarily read at all.
- But most important point is that of course we can always fit model to any dataset with even randomly assigned labels. How you can validate that your trained method really can detect sarcasm?

**Summary Of The Paper:**

In this paper authors propose to use two separate networks, one that is trained to detect hate speech and  the other that is trained to detect sarcasm. The final sarcasm detection score is then a combination of the outputs of both networks.

**Summary Of The Review:**

Quality is not up to the standard.

---

### Decision · Program_Chairs · 2023-01-20

**Decision:**

Reject

**Justification For Why Not Higher Score:**

The authors have not provided any response to the reviewers' comments. Experiments not fully support authors' claims. Poorly written paper.

**Justification For Why Not Lower Score:**

N/A

**Metareview: Summary, Strengths And Weaknesses:**

In this work, the authors  present a technique to  set apart  true hate and  sarcasm speech in texts.   The proposed solution, SarNet, leverages  game theory (Prisoners’ Dilemma) and Nash equilibrium.  SarNet combines ANN and LSTM network to obtain probabilities for the Nash Equalizer, which plays games to identify the label of the sentence, using a quasi-ternary labeling process. The proposed method treats hate and sarcasm as two prisoners.

The problem tackled in this work is of broad interest, and the idea appears to be sound. In particular, all reviewers agreed on the fact that the proposed idea is simple and yet viable. Nonetheless, all reviewers also agreed that the experimental validation is not sufficient to support all of the  authors' claim - one of the reviewers mentioned that the experimental results were not properly analyzed and another stated that "Experiments are basically non-existent." The technical novelty appears to be incremental.

 My opinion is that the authors could have properly address most of the comments, but they decide to not rebut to any of the reviewers' comments. Furthermore, the paper clarity is deficient. All that leaves me with the only choice of proposing to reject the submission.

**Summary Of Ac-Reviewer Meeting:**

N/A